# Knowledge and Practices toward Tuberculosis Case Identification among Accredited Drug Dispensing Outlets Dispensers in Magu District, Northwestern Tanzania

**DOI:** 10.3390/healthcare12020168

**Published:** 2024-01-10

**Authors:** Levina Mwesiga, Stanley Mwita, Deogratius Bintabara, Namanya Basinda

**Affiliations:** 1Department of Health, Christian Social Services Commission (CSSC), Mwanza P.O. Box 905, Tanzania; lmwesiga@cssc.or.tz; 2School of Pharmacy, Catholic University of Health and Allied Sciences, Mwanza P.O. Box 1464, Tanzania; stanleymwita@gmail.com; 3Department of Community Medicine, University of Dodoma, Dodoma P.O. Box 582, Tanzania; bintabaradeo@gmail.com; 4Department of Community Medicine, School of Public Health, Catholic University of Health and Allied Sciences, Mwanza P.O. Box 1464, Tanzania

**Keywords:** accredited drug dispensing outlets, Tuberculosis, TB case identification, Tanzania

## Abstract

Accredited Drug Dispensing Outlets dispensers (ADDO dispensers) have a crucial role in detecting and referring TB suspects. However, several studies highlight low knowledge of TB among ADDO dispensers. To facilitate this, the National TB and Leprosy Control Program trained ADDO dispensers on case identification and referral. Hence, this was a community-based cross-sectional study to determine the knowledge and practice of ADDO dispensers in the detection of active tuberculosis suspects in Magu Districts, Mwanza, Tanzania. This was a cross-sectional study that included 133 systematically selected ADDO dispensers. Out of 133 ADDO dispensers, 88 (66.9%) had attended TB training. About 108 (81%) participants had good knowledge of TB. The majority of ADDO dispensers 104 (78.4%) had poor practice toward the identification of TB cases. Attending training (AOR 4.49, CI 1.03–19.47), longer working experience (AOR 4.64, CI 1.99–10.81), and the presence of national TB guidelines (AOR 3.85, CI 1.11–13.34) was significantly associated with good self-reported TB case identification practices. Therefore, the study revealed adequate knowledge but with poor practice. Provisions to train ADDO dispensers in tuberculosis case detection and referral could yield great results.

## 1. Introduction

Tuberculosis (TB) is one of the causes of morbidity and mortality in the world [1]. It is a major public health problem in Tanzania, leaving the country among the top 30 countries with the highest burden [2,3]. The burden of TB is very high, and its prevalence is 295 cases per 100,000 among adults older than 15 years of age [2,3]. Thus, to achieve better control of TB and minimize patients’ suffering, timely diagnosis and initiation of treatment are critical [4]. This calls for enhanced case finding and collaboration with the private sector to explore innovative undertakings to involve all healthcare providers in improved TB case detection [5]. Local private pharmacy shops are key stakeholders in this work [6,7]. World information show that the majority (between 70% and 95%) of persons living in developing countries consult outlet dispensers for the management of health as the first contact for health care [8]. The reasons include first and foremost the difficulty in accessing different and more expensive kinds of treatment; another is tradition [9]. People would choose these services mainly because they are closer to them, easily affordable, readily available, cheap, and consistent with indigenous cultures or ethnic groups [5,10]. In Tanzania, Accredited Drug Dispensing Outlets (ADDO) dispensers are allowed to offer private services in terms of the provision of drugs to people especially in rural areas [11,12]. ADDO stands for Accredited Drug Dispensing Outlets, which is a program that was established in the year 2003 to improve access to essential medicines and pharmaceutical services to the population living in rural and peri-urban areas [12]. Hence, ADDO dispensers are widely distributed in the country, especially in areas where there are no pharmacies and other health services as well [11,12]. ADDO dispensers’ work is controlled under guidelines as per provisions of the Pharmacy Council of Tanzania [12]. To this end, private and public collaborations for TB case identification and referral are widely recommended [10,13,14,15].

The government encourages essential collaboration in the form of identification of TB cases and referrals [16,17]. Tanzania has moved toward officially recognizing ADDO dispensers as healthcare providers in the country, hence their engagement in TB case identification and referral. Various studies across the world suggest that ADDO dispensers are willing to collaborate with medical practitioners and report referring patients if they are not responding to treatment [6,18,19,20]. Despite their great importance in TB case identification, ADDO dispensers have been depicted as lacking the necessary and sufficient knowledge that could enable their identification of active TB suspects [6,7,20,21,22]. On account of these reports, the National Tuberculosis and Leprosy Programme (NTLP) has provided capacity building to ADDO dispensers to build knowledge and skills to identify persons with TB symptoms and formally refer them to nearby health facilities with diagnostic capacity [17,18,22]. In this way, ADDO dispensers are expected to demonstrate that they are knowledgeable in their practices with patients. To this end, this study aimed to assess the knowledge and practice of ADDO dispensers in identifying and referring active TB suspects in their settings.

## 2. Materials and Methods

### 2.1. Study Design and Setting

This was a community-based cross-sectional study that was conducted at the Magu District Council. It is one of the seven districts of the Mwanza Region of Tanzania, East Africa [23]. The administrative center for Magu District Council is the town of Magu which is found on the side of the Simiyu River. Magu has a lakeshore side on its north, which is bordered by Lake Victoria. However, there is the Busega District Council to the east, the Bariadi Municipal Council to the south by Itilima District, Maswa District, Kwimba District, and Misungwi District, and to the west by the city of Mwanza, which consists of Nyamagana District and Ilemela District. The district population under the Magu District Council is 299,759 [24]. The district has around 200 ADDO dispensers, as per the 2020 report of the MoHCDGEC. Magu is one of the rural districts where capacity building on TB identification and referral training was completed among ADDO dispensers.

### 2.2. Study Population, Sample, and Sampling Procedure

The study participants included all registered ADDO dispensers present in the Magu District Council. Only ADDO dispensers who are registered by the Pharmacy Council of Tanzania and are on the list of district-accredited drug dispensing outlets dispensers were included in the study. No effort was made to recruit ADDO dispensers who were not at the station of work during the time of the study.

The sample size was 133, as determined by the Taro Yamane formula based on the 200 registered ADDO dispensers in Magu [25]. Systematic random sampling was used to obtain the sample size required for the study [26]. Under this sampling technique, a list of registered names of ADDO dispensers was obtained from the District Pharmacist officer and arranged alphabetically. After selecting the first ADDO dispenser by rotary, every 2nd (Kth value) was selected until the required sample was obtained. Participants who were selected were screened for eligibility criteria during fieldwork. In case where one participant had no eligible criteria, he or she was replaced by another participant who was selected purposefully to replace the non-eligible participant.

### 2.3. Data Collection and Analysis

Data were collected using a standardized questionnaire developed from questionnaires (Appendix A) that had been used in previous studies and from various articles and books on information related to ADDO dispensers and their scope of work about TB [18,27,28,29]. The questionnaire was reviewed by three expert who looked at the contents of the questions and made recommendations. The questionnaire was developed in English and then translated into Swahili. It was back-translated by another person conversant in both languages to ensure no loss of translation. The questionnaire was pretested in Nyamagana district to sort out wording issues or questions that were difficult to answer; thus, the questionnaire was examined to see if it can measure what it was meant to measure. Trained research assistants collected data.

Correct answers on knowledge cumulatively gave a 100% score. The mean score for knowledge was calculated, and it was 75%. Hence, in our study, those who scored 75% and above were considered to have adequate knowledge, and those who scored below had inadequate knowledge. The same was seen for practice, those who scored less than the mean average were categorized as having inadequate self-reported practice and vice versa. The scoring was completed as per recommendations for designing KAP studies [30].

The collected data were coded, entered, cleaned, and analyzed using the Statistical Package for Social Studies (SPSS) version 26 (IBM Corp, Armonk, NY, USA). Demographic characteristics were summarized as means and standard deviations for normally distributed continuous variables, and medians and interquartile range were used for skewed continuous variables. Frequencies, percentages, and proportions for categorical variables. Bivariate logistic regression was first performed for each independent variable. Variables that were significant in bivariate analysis (*p*-value less than 0.05) were taken to multivariate analysis. A multivariable linear regression analysis was completed to identify the factors associated with the correct active TB case identification and referral practice of ADDO dispensers after controlling for confounding variables. Assumptions for binary logistic regression, multicollinearity, and outliers were considered and assessed before fitting and were combined linearly when being added to the multivariable linear regression. Any variables whose *p*-values were less than 0.05 were considered statistically significant.

## 3. Results

### 3.1. Socio-Demographics Characteristics of the Study Participants

A total of 133 study participants were enrolled in the study. Of these, 99 (74.5%) and 34 (25.5%) were female and male, respectively. Participants’ age distribution was as follows: 41 (30.7%) were between 18 and 25 years, 67 (50.2%) were between 26 and 35 years, 23 (17.2%) were between 36 and 45, and 3 (1.9%) are >45 years old. The mean age of the participants was 27.6 (±4.7 SD) years. The majority 97 (72.9%) of the respondents were married. Most of them 123 (92.5%) had a certificate level of education. About 85 (63.90%) have been working as an ADDO dispenser for less than one year. The mean duration of working as an ADDO dispenser was 15 (±3.9 SD) months. Around 89 (67%) participants reported having nearby health facilities around their areas with a mean of 28 min ± 9 SD to reach the nearby health facility; see Table 1.

### 3.2. ADDO Dispensers Knowledge of TB

All 133 (100%) study participants had heard about TB. This is shown in Table 2. Almost all (91, 68.4%) participants obtained information on TB from multiple sources of information, the rest (42, 31.6%) obtained such information from either friends, health workers, health extension workers, media, patients, family members, reading books, or via seminars or conferences. About 88 (65.9%) of them knew that bacteria are the causative agent of TB, 32 (24%) said that is caused by a virus, and the rest (13, 10.1%) stated another source of infection apart from bacterial or virus.

However, the common signs and symptoms of TB mentioned by the participants include cough (100, 75%), cough for 2 or more weeks (129, 98%), cough up blood (101, 76%), weight loss of (124, 93.2%), shortness of breath (19, 14.3%), fever or sweat at night (118, 89%), and chest pain (21, 15.8%). All the participants knew that TB can be transmitted from one person to another. Inhaled droplets through coughing and sneezing were recognized as the common source of TB infection by 106 (79.9%) of the respondents. Still, exposure to dust (87, 65.4%), exposure to cold air (83, 62.6%), and drinking raw milk (59, 44.8%) were also mentioned as important modes of transmission. When asked who they think can transmit TB, 122 (92%) reported that persons who have active TB and are not taking TB drugs, 76 (56.9%) mentioned anyone who has a cough, and 57 (42.6%) answered that any person who is diagnosed with TB and has finished treatment.

All participants in this study agreed that the transmission of Pulmonary TB (PTB) can be prevented. Of the respondents, 88 (66%) considered covering their mouth and nose as the most commonly used method for preventing the spread and transmission of TB.

Moreover, 71 (53.3%) respondents mentioned that the transmission and spread of TB could be prevented by closing windows (61, 46%), avoiding sharing cups (61, 46.2%) and separating dishes (38, 28.9%). Regarding who can be infected with TB infections, 120 (90.3%) reported that anyone may be infected, 52% said HIV-infected people, 33.8% reported poor people and 47% said that alcoholics are the ones infected by TB. The majority of participants, 102 (76.8%), believed that TB is a curable disease. Similarly, 88 (66.1%) respondents stated that the disease was curable with specific drugs given by health facilities, but 45 (33.8%) of respondents did not know how it is cured. Only 43.1% of respondents knew the current drugs for TB treatment; 50.5% did not know.

On scoring knowledge, more than two-thirds 108 (81%) of participants had adequate knowledge of TB. A few of them 25 (19%) had inadequate knowledge of TB.

### 3.3. Presumptive TB Case Identification and Referral Practice among ADDO Dispensers

In the current study, participants reported attending an average number of 20 ± 6 SD customers per day, and only a few participants 12 (9.02%) do keep records of the signs and symptoms of the customers they attend. Other services apart from dispensing drugs that were mentioned to be offered by the participants to their clients include counseling (79, 59.5%), provision of health education (32, 24%), referring to nearby health facilities (4, 3%), and offering multiple services to their clients (18, 13.5%). During the study period, more than half of the participants (98, 73.9%) reported having ever seen a person(s) with signs and symptoms of TB in the last year, although only 59 (44%) agreed to having ever seen a person with signs and symptoms of TB in the last year. Others (23, 17%) reported having attended to such patients more than a year ago; see Table 3.

The majority of patients came with a combination of symptoms that made them suspected TB cases. The most frequently reported symptoms were chronic cough (115, 86.5%), production of sputum (104, 78.3%), night sweats (86, 64.3%), fatigue or tiredness (84, 63.3%), shortness of breath (75, 56.5%), and unexplained weight loss (67, 50%). Most (93, 70.1%) of the participants who suspected TB cases provided health education on TB, and the client left. Very few (6, 4.2%) participants provided anti-TB drugs and referred them to the nearby health facility, 26 (35%) referred their clients to the nearby health facility, and 8 (6.01%) had multi-reactions.

Less than one-quarter (21.6%; 95% CI 16.2% to 23.8%) of ADDO dispensers scored equal to or above the mean score, which was categorized as good practice for the identification of TB cases. Thus, the majority (78.4%) in this study had poor practice in the identification of TB cases.

### 3.4. Factors Associated with TB Knowledge among ADDO Dispensers

In the bivariate analysis, male sex, duration of working as an ADDO dispenser, educational status, distance of nearby health facility, presence of national TB guidelines or any other academic materials, and having ever attended any TB training were significantly associated with knowledge of respondents about TB. However, in the stepwise multivariate analysis, educational level, duration of working as an ADDO dispenser, and attending any TB training remained significantly associated with the knowledge of respondents about TB after controlling other factors. Accordingly, ADDO dispensers who had a postgraduate degree were more likely to have good knowledge about TB than ADDO dispensers who had a diploma (AOR) = 5.78; 95% CI 2.3 to 14.33). ADDO dispensers who had ever attended any TB training were more likely to have good knowledge about TB than those who did not receive any form of training (AOR = 1.79; 95% CI 1.00–3.17). Similarly, ADDO dispensers who had been working for more than a year were more likely to have good knowledge about TB than those who had been working for less than one year (AOR = 1.85; 95% CI 1.12–3.03). See Table 4

### 3.5. Factors Associated with TB Case Identification and Referral among ADDO Dispensers

In the bivariate analysis, working site, age, duration of working as an ADDO dispenser, educational status, presence of national TB guidelines or any other academic materials, knowledge on TB, having ever attended any TB training, having ever attended a person with signs and symptoms of TB in the last year and tendency of facing any challenge in identifying the presumptive TB case were associated with TB case identification practices among ADDO dispensers. However, in the multivariate analysis, the age of respondents, duration of working as an ADDO dispensers, presence of national TB guidelines or any other academic materials, knowledge of TB, having ever attended any TB training, having ever attended a person with signs and symptoms of TB and tendency of facing any challenge in identifying the presumptive TB case were significantly associated with TB case identification practices among Accredited Drug Dispensing Outlets dispensers. ADDO dispensers who worked more than 1 year (AOR = 4.64; 95% CI 1.99–10.81) were more likely to have good self-reported practice on TB case identification than those who had been working for less than a year. ADDO dispensers in the age group of 26–35 years (AOR = 2.70; 95% CI 1.27–5.76) and above 35 years (AOR = 4.42; 95% CI 1.77–11.08) were more likely to have good self-reported practice on TB case identification as compared with those ADDO dispensers with the age group of <25 years. ADDO dispensers with good knowledge about TB were more likely to have good self-reported practice than those with poor knowledge (AOR = 4.49; 95% CI 1.03–19.47). ADDO dispensers who were supplied with national TB guidelines or any other academic materials were more likely to have good self-reported practice on TB case identification as compared with those who had no such materials (AOR = 3.85; 95% CI 1.11–13.34). Those who had ever attended any TB training were more likely to have good self-reported practice on TB case identification than those who never attended (AOR = 4.49; 95% CI 1.03–19.47). See Table 5. 

## 4. Discussion

ADDO dispensers are the principal source of medicines in Tanzania as well as important suppliers of healthcare services in the community [22]. This aligns with many studies that have shown the critical role that retail drug sellers play in the health systems of many resource-limited countries [22,31]. While the deficit of literature regarding their role in the identification of TB cases in their setting keeps on increasing daily, this study investigated their level of knowledge about TB, determined their practice in identifying TB suspects, and explored the factors that affect their level of knowledge and practices toward identifying TB cases in their areas.

The current study has found the overall knowledge of ADDO dispensers toward TB to be good (81%). The findings of this study are in line with the findings of the studies conducted in Saudi Arabia, Uganda, and other places that showed similar cadre workers to have adequate knowledge regarding TB [28,32,33,34]. The high level of knowledge might reflect the impact of training ADDO dispensers that have been conducted by the Pharmacy Council of Tanzania, and this has helped ADDO dispensers stay updated with current NTLP guidelines in low-resource settings [21].

Most of the respondents knew that TB is caused by bacteria, as well as TB signs and symptoms, including the four basic TB symptoms (cough, fever, weight loss or not gaining weight, and night sweats). Furthermore, our study has revealed that all the participants knew that TB can be transmitted from one person to another. Inhaled droplets through coughing and sneezing were recognized as the common source of TB infection. Also mentioned are exposure to dust, exposure to cold, and drinking raw milk.

On the other hand, our study found ADDO dispensers to have had generally poor self-reported TB case identification practices regarding TB referral and its management. The majority (78.4%) had poor practice in the identification of TB cases. This is similar to previous studies both in Tanzania and elsewhere [13,18,21,35]. Inadequate TB identification practice among ADDO dispensers and other healthcare workers (HCWs) may result in substandard care, ineffective service provision, inefficient resource use, and an impact on health outcomes as well as an increase in the risk of TB transmission and resistance development.

In the current study, it is reported that non-referral was attributed to a negative attitude put up by the customers who visit their shops. This is partly due to their refusal to report to the designated health facility and also because they were not open enough to allow the shop operators to talk to them. This finding is similar to other studies, yet some participants also think they may lose customers to other ADDO dispensers [29,36,37]. A lack of formal referral and feedback loops has been shown to discourage ADDO dispensers from referring clients. Other studies also show that healthcare workers may not be welcoming enough to clients who are referred to them, hence putting off the clients and the referring ADDO dispensers [38,39,40,41].

In this study, a number of factors were associated with better knowledge of TB among ADDO dispensers. They include educational level, duration of working as an ADDO dispenser (working experience), and history of attending TB training. These factors are similar to different studies conducted elsewhere affecting the knowledge of TB [18,28,31,32,33,41]. However, training on TB has been highlighted to be a very important factor in the practice of case identification and referral. All these studies suggest lack of training is a big obstacle to TB case identification and subsequent referral.

Our study has some limitations. While we tried to enroll all ADDO dispensers from villages of Magu, the sample size represents a small proportion of the thousands of ADDO dispensers deployed to work in Mwanza and Tanzania at large each year. Furthermore, our study measured knowledge and self-reported practice, and we did not measure attitude as an important factor. This is because we wanted to understand after training whether the ADDO dispenser would know and practice TB case identification and referral. Therefore, even after benchmarking our tool against other similar studies at national and international levels and further validation in terms of its face and content, the differences we are observing in this study could be a result of the variability in the KAP definition.

Notwithstanding the above limitations, this is the first attempt to explore the level of ADDO dispensers toward TB and determine their practice in TB case identification in the Mwanza region. The identified knowledge gaps and poor practices can serve as baseline data to design effective interventions to address these shortcomings. Further studies at larger scales including qualitative methods, and the observation of actual practices as well as considering the availability of resources and other possible infrastructure constraints are warranted.

## 5. Conclusions

The findings of this study have revealed adequate knowledge of TB, however, with poor related practice of identifying TB cases and referring them to advanced treatment. Training on TB has been highlighted as a significant factor in improving both knowledge and practice. To this end, periodic training and making guidelines available could be vital to improve the detection and referral of TB. This study assessed self-reported practice, and hence, a study to observe the actual ADDO dispensers practice is recommended.

## Figures and Tables

**Table 1 healthcare-12-00168-t001:** Socio-demographic characteristics of respondents.

Variable	Response	Frequency	Percent
Sex	Male	34	25.6
	Female	99	74.4
Age	18–25	41	30.8
	26–35	67	50.4
	Above 35	26	19.6
Marital Status	Single	33	24.8
	Married	97	72.9
	Separated	2	1.5
	Divorced	1	0.75
Highest of Education	Certificate	123	92.5
	Diploma and above	10	7.5
Duration of working as an ADDO * dispenser	Less than 1 year	85	63.9
	More than 1 year	48	36.1
Presence of nearby health facilities in this area	Yes	89	66.9
	No	44	33.1
How long does it take to reach the nearby health facility	1–15 min	21	15.8
	16–30 min	82	61.7
	30 min–1 h	11	8.3

* Accredited Drug Dispensing Outlets.

**Table 2 healthcare-12-00168-t002:** ADDO dispensers’ knowledge of TB.

Variable	Response	Frequency	Percent
Have you ever heard about TB *?	Yes	133	100
	No	0	0
Source of information about TB	Friend	2	1.5
	Health workers/Health extension workers	9	6.8
	Media (radio, TV ** posters, etc.)	10	7.5
	Patient	4	3.0
	family members/friends	1	0.75
	Via reading books	6	4.5
	Via seminar/conferences	21	15.8
	Multiple sources	91	68.4
What do you think is the cause of PTB ***	Bacteria/germ	88	66.2
	Virus	32	24.1
	Fungus	5	3.8
	Witchcraft	2	1.5
	poverty	2	1.5
	Living together with an untreated TB patient	1	0.75
	Cold air	1	0.75
	Unventilated house	2	1.5
What are some of the common symptoms of TB	Cough	100	75.2
	Cough for 2 or more weeks	129	97
	Cough up blood	101	75.9
	Weight loss	124	93.2
	Shortness of breath	19	19.3
	Chest pain	21	15.8
	Fever/sweat at night	118	88.7
Do you know that TB can be transmitted	Yes	100	75.2
	No	0	0
How can TB be transmitted from person to person	Through droplets from coughing or sneezing of a person having TB	106	79.7
	Drinking raw milk	59	44
	Exposure to dust	87	65.4
	Exposure to cold air	83	62.4
Who do you think can transmit TB	Anyone who has a cough	76	57.1
	A person who has active TB and not taking TB drugs	122	91.7
	A person who is diagnosed having TB and finished treatment	57	42.9
Do you think that the transmission of PTB can be prevented	Yes	100	75.2
	No	0	0
How can TB be prevented	Covering mouth and nose while coughing or sneezing	88	66.2
	Avoiding sharing cups	61	45.9
	Separating dishes	38	28
	Through closing windows	71	71.4
In your opinion, who can be infected with TB	Anybody	120	90.2
	Poor people	45	33.8
	Homeless people	0	0
	Only alcoholics	63	47
	Only people living with HIV	69	52
Can TB be cured	Yes	102	76.7
	No	31	23.3
How can someone with TB be cured	Traditional medicine	0	0
	Home rest without medicine	0	0
	Praying	0	0
	Specific drugs given by health facility	88	66.2
	Dots	0	0
	Do not know	45	33.8
Do you know the drugs for TB	Yes	57	42.9
	I don’t know	76	57.1

* Tuberculosis, ** Television, *** Pulmonary Tuberculosis.

**Table 3 healthcare-12-00168-t003:** Presumptive TB case identification and referral practice of ADDO dispensers.

Variable	Response	Frequency	Percent
Do you keep records of the signs and symptoms of the customers you attend?	Yes	12	9.02
	No	121	91
Other services, apart from dispensing drugs, you offer to your clients	Counseling	79	59
	Provision of health education	32	24.1
	Refer to a nearby health facility	4	3
	Offer multiple services	18	13.5
Have you ever seen a person(s) with signs and symptoms of TB at your shop in the last year?	Yes	98	73.9
	No	35	26.1
Have you ever seen a person with signs and symptoms of TB in the last year?	Yes	23	17
	No	110	83
What signs and symptoms did you consider to suspect that your customer was a TB case	Cough	115	86.5
	Cough for 2 or more weeks	129	97
	Cough up blood	104	78.3
	Weight loss	67	50
	Shortness of breath	75	56.5
	Chest pain	21	15.8
	Fever/sweat at night	86	64.3
What service did you offer to a TB case identified	Provided health education on TB and the client left	93	70.1
	Referred their clients to the nearby health facility	32	39.2
	Multiple services	8	6.01
Do you tend to refer persons with symptoms of TB/seriously sick to advanced treatment?	Yes	32	24
	No	101	76
If yes, where do you refer to advanced treatment N = 32	Nearby health facilities	28	87.5
	Other specialists	4	12.5

**Table 4 healthcare-12-00168-t004:** Factors associated with TB knowledge among ADDO dispensers.

Variables	Category	TB Knowledge		OR (95% CI)	AOR (95% CI)	*p*-Value
		Adequate	Inadequate			
Age	18–25	39	12	1.00	1.00	
	26–35	54	13	1.17 (0.74–1.85)	0.11 (0.09–1.72)	0.301
	35+	24	9	0.77 (0.40–1.49)	0.32 (0.40–3.17)	0.348
Sex	Female	25	8	1.00	1.00	
	Male	82	17	1.81 (1.17–2.80)	0.09 (0.05–0.15)	0.855
Distance to a health facility	More than 1 h	12	7	1.00	1.00	
	30 min–1 h	9	2	0.17 (0.07–0.40)	1.04 (1.00–2.71)	0.059
	16–30 min	69	13	0.18 (0.08–0.39)	0.95 (0.43–1.34)	0.553
	Less than 15 min	12	9	0.22 (0.11–0.41)	2.54 (1.00–6.54)	<0.001
Presence of TB guidelines	No	12	93	1.00	1.00	
	Yes	20	8	3.00 (1.68–5.44)	0.33 (0.50–2.71)	0.015
Work experience	Less than 1 year	22	63	1.00	1.00	
	More than 1 year	39	9	2.68 (1.69–4.25)	1.85 (1.12–3.03)	0.0019
Ever attended any TB training	No	44	172	1.00	1.00	
	Yes	89	55	1.78 (1.04–3.05)	1.79 (1.00–3.17)	0.002

AOR: Adjusted Odds Ratio, CI: Confidence Interval.

**Table 5 healthcare-12-00168-t005:** Factors associated with TB case identification and referral among ADDO dispensers.

Variables	Category	Practice		OR (95% CI)	AOR (95% CI)	*p*-Value
		Good	Poor			
Age	18–25	19	22	1.00	1.00	
	26–35	18	49	2.65 (1.36–5.16)	2.70 (1.27–5.76)	0.055
	35+	10	16	4.05 (1.82–9.00)	4.42 (1.77–11.00)	0.301
Sex	Male	7	16	1.00	1.00	
	Female	17	82	0.47 (0.28–0.78)	0.57 (0.232–1.411)	0.225
Duration of work	Less than 1 year	32	53	1.00	1.00	
	More than 1 year	12	36	3.21 (1.71–6.04)	4.64 (1.99–10.81)	0.014
Education level	Certificate	31	92	1.00	1	
	Diploma	8	2	0.31 (0.12–0.81)	1.89 (0.78–2.10)	0.075
Ever attended any TB training	No	33	11	1.00	1.00	
	Yes	72	27	2.04 (0.90–4.60)	4.49 (1.03–19.47)	<0.001
Ever seen a person with signs and symptoms of TB	No	12	89	1.00	1.00	
	Yes	14	18	1.07 (0.44–2.88)	5.21 (0.86–31.51)	0.0042
Supplied With TB guideline	No	36	53	1.00	1.00	
	Yes	10	34	0.59 (0.21–1.63)	3.85 (1.11–13.34)	<0.001
Face challenge in identifying presumptive TB case	No	19	8	1.00	1.00	
	Yes	12	194	1.88 (1.29–3.57)	3.81 (2.11–4.28)	0.897
Knowledge of TB	Poor	9	16	1.00	1.00	
	Good	19	6	2.7 (1.6478–19.2329)	5.6296 (0.3942–2.004)	<0.0058

AOR: Adjusted Odds Ratio, CI: Confidence Interval.

## Data Availability

All data and information within this manuscript are in the form of tables and other details.

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
