# Peer review of "Knowledge and Practices toward Tuberculosis Case Identification among Accredited Drug Dispensing Outlets Dispensers in Magu District, Northwestern Tanzania"

_healthcare, 2024, doi:10.3390/healthcare12020168_

Round 1
Reviewer 1 Report
Comments and Suggestions for Authors
Dear Respectable Authors
Thank you for considering a great area of research related to public health issues. Investigating the KAP is a routine method in public health but conducting such research has some limitations and barriers. You investigated the knowledge and practice of ADDOs in the detection of active tuberculosis suspects in Magu Districts-Mwanza, Tanzania. Your results are interesting but your manuscript needs some modifications as follows;
- Line 18, in my opinion, the correct is deviation not variations. Please clear it.
- From the scientific point of view, the number of your samples is small, but considering the things you have mentioned and the limited number of these people, this sample size is acceptable. However, this should be mentioned in the weakness section.
- Usually, such studies are known as KAPs. Why did you miss the attitude part? Perhaps one of the reasons that people have high knowledge but poor performance is related to the type of attitude that needs to be investigated. This is another weakness of your study.
- Line 86, you do not mention eligibility criteria. Please add these criteria.
- Please add some information regarding the validity and reliability of the questionnaire.
- In what language is the questionnaire prepared? And in what language is it distributed?
- Line 95 does not match the line 17. 25 or 26?
- Line 96, "variations" is not correct.
- Line 118, table caption, What you mean by " This is a table. Tables should be placed in the main text near to the first time they are cited." Please remove this statement and insert the correct title for your table.
- Please attach your questionnaire as a supplementary file to the method section, line 93.
- Table 2, column 2, please remove the word "from" in this column.
- Please add a section for "strengths, weaknesses, and limitations" of the study.
- Lines 277-8, you mentioned several tables and figures as supplementary files. I have no access to these supplementary files and also these files are not referenced or mentioned in the text.
Comments on the Quality of English Language
There are several punctuation and grammatical errors throughout the text. For example, in line 80, "basing" is not correct, please replace it with "based".
Line 93, the end of the sentence is open. Please add a point.
Line 20, please add an inter after 99.
Author Response
Dear Reviewer,
We are thankful for your reviews. We have worked on them, one by one. Please see the attached document.
Let us know if you have further comments or questions.
Regards

Reviewer 2 Report
Comments and Suggestions for Authors
Thank you for inviting me to review this work entitled "Knowledge and Practices towards Tuberculosis Case Identification among Accredited Drug Dispensing Outlets Dispensers in Magu District, Northwestern, Tanzania" which aimed to to determine the knowledge and practice of ADDOs dispensers in the detection of active tuberculosis suspects in Magu Districts-Mwanza, Tanzania.
Actually, it is a very interesting study and I believe this concept of having Accredited Drug Dispensing Outlets Dispensers (ADDOs) is not well-known for healthcare professionals and deserves to be highlighted.
Some comments were raised and need to be addressed:
General comment
· English language proof read is required.
Abstract:
· Please follow the authors guidelines regarding the abstract, it should be structured.
· Please clarify the recruitment method in the abstract. Did you conduct structured interviews using a Questionnaire?
· Rewrite this sentence comprehensively " Data was analyzed using SPSS version 25 with numerical and categorical data summarized as means and standard variations and frequencies and percentages respectively. A P-value of less than 0.05 was considered significant for association in logistic regression." And focus on the main used method and Questionnaire administration and development.
· Rewrite this sentence comprehensively " Out of 133 ADDOs, 99(74.5%) were females and 34 (25.5%) were males with the mean age of the 20 participants being 27.6 (±4.7 SD) years." We used to write one of the genders.
· Did you calculate knowledge score; hence, you mentioned a good knowledge? Then at the end, you said adequate knowledge?
Introduction
· Please write more about Accredited Drug Dispensing Outlets Dispensers (ADDOs), and write the acronym first time mentioned in the introduction.
· Research gap is not clear enough. Please elaborate.
· Clarify the sample size calculations.
· Regarding "Systematic random sampling was used to get the sample size required for the study." Please add a reference for the sampling technique
· Please add details about the study tool composition, development, validation and data recruitment.
· Knowledge scoring and calculations?
Results:
I believe Table 2 does not all questions represent knowledge, hence, there are many questions that appear perception or attitude! Unless there is a reference for this classification.
Comments on the Quality of English Language
It needs English language editing and proofreading
Author Response
Dear Reviewer,
Thank you for your comments and advice to improve the work.
We have addressed your comments, one by one, and improved the work. Please feel free to let us know if you have any further comments.

Reviewer 3 Report
Comments and Suggestions for Authors
This is an interesting study evaluating the knowledge and practice of ADDOs in a region in Tanzania regarding TB. Several major and minor comments need to be addressed as below:
1. Lines 17-19: I suggest removing the statistical analysis part as this is uncommon to be mentioned in the abstract. Instead, the authors are encouraged to add more to the background. For example, provide a brief background on how significant the issue of TB in Tanzania.
2. Line 29: Remove the numbers from the keywords. I also suggest adding "Tanzania" as one of the keywords.
3. Introduction: Provide a brief background on the current situation of TB in Tanzania. For example, it is endemic in the country, latest statistics on the number of cases… etc.
4. Line 69-71: If possible, please provide more recent population data that are not more than 10 years old.
5. Line 75: You may use the abbreviation ADDOs here since you have already spelled it out previously.
6. Methods (2.3 Data collection and analysis): Please clarify if the responses were collected via in-person interviews with the participants or were the questionnaires distributed online?
7. Methods (2.3 Data collection and analysis): Add a brief statement describing the questionnaire: How many questions it contained? Was it developed based on previous studies or was it developed by experts? Was it piloted (i.e., tested on a small sample for validation purposes and potential editing of questions if needed)?
8. Line 95: Add the name of the software developer and their region (IBM Corp, Armonk, NY, USA).
9. Lines 99-100: What was the p-value criterion that you used to include variables in the multivariate regression? Was it a p<0.2, <0.1< or <0.05? Or did you include all the factors without exception? Please clarify.
10. Results (3.1.): This first paragraph largely replicates the data from Table 1. I suggest shortening it. For example, lines 109-111 on the details of age distribution could be deleted.
11. Line 118: Please provide an appropriate title for the table. Also, please provide a spell out to the abbreviaion ADDO in the footnote.
12. Lines 130-140 (minor comment): Please use small cases of the letters.
13. Line 141: Please spell out PTB as pulmonary TB, then put the abbreviation PTB in between parentheses.
14. Lines 153-154: What is your definition of adequate vs. inadequate knowledge? What were the scores cutoffs? This definition should be added to the methods section.
15. Table 2: The first 2 variables should be further divided. For example, "Ever heard about the disease called TB" should have Yes/No. And then another variable below it should be titled "Where you heard about TB", which should corresponds to the responses "From friend, From health workers, etc." Corresponding to "Bacteria/germ, Virus, Fungus", a variable titled "Cause of TB" is suggested to be added and separated from the previous variable of "Source of information about TB."
16. In the footnote of Table 2, please spell out the abbreviation PTB.
17. Lines 175-178: Again, please provide a definition to what you call "good practice for identification of TB cases" in the methods section.
18. Table 3. There are 3 stars next to "What signs and symptoms did you consider to suspect that your customer was a TB case." Please indicate in the footnote what this mark mean. Otherwise, please delete it.
19. Line 182: Add "male" before "sex." Also, age was not significant (95% CI crosses 1).
20. Table 4: Replace COR with just OR.
21. Table 4: What does the column "TB knowledge" mean? Do you mean % knowledgeable of TB? What each subcolumn numbers indicate? Please add a clarification below the "TB Knowledge" title. Also, I suggest replacing "Duration of working" with "Work experience."
22. Lines 189-190: This variable (educational degree) wasn't listed in Table 4. Please add it.
23. Line 209: Replace the spell out "Accredited Drug Dispensing Outlets" with the abbreviation "ADDOs"
24. Table 5 The 4th column should be titled OR (95% CI). It currently only has 95% CI.
25. Table 5: The OR and AOR, as well as their 95% CI are the exact same number for the factors: Ever attended any TB training and Knowledge of TB, which is questionable. Please double check all the numbers and that they were copied correctly from SPSS.
26. Lines 199-204: Some of the factors listed were not significant in the bivariate analysis (95% crosses 1) and should be removed from this sentence: Ever attended any TB training, Ever seen a person with signs and symptoms of TB, Supplied With TB Guideline, and Knowledge of TB.
27. Line 246: Please add that those factors were incorrectly chosen.
28. Line 251: Spell out HCWs.
29. The paper lacks a study limitations paragraph at the end of the discussion section and before the conclusion. If this questionnaire was just developed for this study and not previously validated, then this should be mentioned as one of the limitations. The limited geographic area of the study may limit its generalizability to other regions of Tanzania for example. The authors may also think of other potential limitations to be included in this paragraph.
30. Conclusion: You also noticed that the presence of national TB guidelines was significantly associated with better knowledge and practice of TB identification. Therefore, authors should also emphasize the importance of developing national TB guidelines and include them in the training sessions offered to ADDOs.
Author Response
Dear Reviewer,
We thank you for taking the time to look at our work.
We have worked on all the comments given (one by one).
Please feel free to let us know if you have other comments.

Reviewer 4 Report
Comments and Suggestions for Authors
Thank you for your submission. It is quite interesting.
It is unclear how the questionnaire was administered.
The sample size was calculated as 133, yet a target of 146 was later mentioned without any context given. This is unclear.
What exactly does this statement mean? “A researcher systematically selected the participants using a well-structured format until the required sample was obtained”. How can other researchers replicate this?
What is the relevance of the religion of the participants?
It would help readers from other countries to have more information on the training requirements to become a dispenser in Tanzania.
The statistical analyses are incorrect and too simplistic. Rather than arbitrarily splitting the participants into two groups for the analyses, especially without any validation of the cutoff scores selected, the statistical analyses should use the numerical values of the scores as continuous numerical variables and multiple linear regression (not logistic regression) should be used.
We are not told how it was decided to include variables from the bivariate analyses into the subsequent multivariate analyses. It is also unclear how and whether the issue of multicollinearity between variables was addressed. It is quite possible, for instance, that the better educated dispensers were more likely to attend educational events. One would also expect collinearity between age and duration of working as a dispenser.
The English needs careful revision.
Comments on the Quality of English LanguageThe English needs careful revision, including the appropriate use of tenses.
Author Response
Dear Reviewer,
Thank you for the time to look at our work.
We have addressed your comments one by one.
Please let us know if you have any further comments.

Round 2
Reviewer 2 Report
Comments and Suggestions for Authors
Many thanks for the author for their efforts to address the comments
Author Response
We thank you for your review and are grateful for your comments which have significantly improved our work.
Reviewer 3 Report
Comments and Suggestions for Authors
Thank you to the authors for improving the manuscript. I have no further comments.
Author Response
Thank you for your review that have significantly improved our work. Kind regards.
Reviewer 4 Report
Comments and Suggestions for Authors
Thank you for your revision. The manuscript is improved.
As before, however, the statistical analyses are incorrect and too simplistic. Rather than arbitrarily splitting the participants into two groups for the analyses, especially without any validation of the cutoff scores selected, the statistical analyses must use the numerical values of the scores as continuous numerical variables and multiple linear regression (not logistic regression) must be used. Please consult a statistician.
It is also unclear how and whether the issue of multicollinearity between variables was addressed. It is quite possible, for instance, that the better educated dispensers were more likely to attend educational events. One would also expect obvious collinearity between age and duration of working as a dispenser. This has not been addressed at all.
There are still some remaining English issues.
Comments on the Quality of English LanguageOnly a few remaining English issues.
Author Response
Dear Reviewer,
We are thankful for your critical review. We have addressed your comments and incorporated them into the text. We are open to more reviews, and we will work hard to improve the quality of the work.
Kind Regards
